# All-printed nanomembrane wireless bioelectronics using a biocompatible solderable graphene for multimodal human-machine interfaces

Young-Tae Kwon[1,6], Yun-Soung Kim[1,6], Shinjae Kwon[1], Musa Mahmood[1], Hyo-Ryoung Lim[1], Si-Woo Park[2], Sung-Oong Kang[2], Jeongmoon J. Choi[3], Robert Herbert[1], Young C. Jang [3,4], Yong-Ho Choa [2] & Woon-Hong Yeo [1,4,5✉]

Recent advances in nanomaterials and nano-microfabrication have enabled the development of flexible wearable electronics. However, existing manufacturing methods still rely on a multi-step, error-prone complex process that requires a costly cleanroom facility. Here, we report a new class of additive nanomanufacturing of functional materials that enables a wireless, multilayered, seamlessly interconnected, and flexible hybrid electronic system. All-printed electronics, incorporating machine learning, offers multi-class and versatile human-machine interfaces. One of the key technological advancements is the use of a functionalized conductive graphene with enhanced biocompatibility, anti-oxidation, and solderability, which allows a wireless flexible circuit. The high-aspect ratio graphene offers gel-free, high-fidelity recording of muscle activities. The performance of the printed electronics is demonstrated by using real-time control of external systems via electromyograms. Anatomical study with deep learning-embedded electrophysiology mapping allows for an optimal selection of three channels to capture all finger motions with an accuracy of about 99% for seven classes.

[1] George W. Woodruff School of Mechanical Engineering, Institute for Electronics and Nanotechnology, Georgia Institute of Technology, Atlanta, GA 30332, USA. [2] Department of Materials Science and Chemical Engineering, Hanyang University, Ansan 15588, South Korea. [3] School of Biological Sciences, Georgia Institute of Technology, Atlanta, GA 30332, USA. [4] Wallace H. Coulter Department of Biomedical Engineering, Parker H. Petit Institute for Bioengineering and Biosciences, Georgia Institute of Technology, Atlanta, GA 30332, USA. [5] Neural Engineering Center, Flexible and Wearable Electronics Advanced Research, Institute for Materials, Institute for Robotics and Intelligent Machines, Georgia Institute of Technology, Atlanta, GA 30332, USA. [6]These authors contributed equally: Young-Tae Kwon, Yun-Soung Kim. ✉email: whyeo@gatech.edu

Recent progress in the state-of-the-art wearable electronics describes the merits of the thin and stretchable hybrid electronic packages toward improved wearability and ultimately the seamless integration with user's body and lifestyle[1,2]. However, the route to manufacturing such electronics typically requires microfabrication processes that are inevitably wasteful, cost-prohibitive, and not scalable. Notable examples are the recent breakthroughs in the fabrication of thin, multilayered flexible electronics with abilities to mechanically conform to human physiology as well as incorporate commercial electronic components for functional bioelectronic application[3–5]. While these platforms successfully demonstrated the utility of the well-defined, traditional CMOS processes toward unique wearable applications, the fabrication processes require the access to cleanroom facility, high-vacuum equipment, and dedicated personnel for maintenance. From this point of view, the ability to manufacture stretchable hybrid electronics entirely based on additive manufacturing methods is particularly attractive due to decreased material consumption, fast turnaround, scalable fabrication based on parallel printing, and, most importantly, the fact that only a single equipment is needed[6]. With advances in novel printing methods and soft materials, wearable electronics are transitioning from rigid modalities based on metals and plastics to soft form factors, which offer comfortable, seamless integration with the skin[7,8]. Development of highly conductive nanomaterials and annealing methods of printed inks, including Ag[9,10], Cu[11], and carbon nanotube materials (CNT)[12] enable low skin-to-electrode contact impedance, leading to improved signal-to-noise ratios in electrophysiological recordings during dynamic body movements. Leveraging these advances, several printed wearable systems have been demonstrated, limited only to passive electrodes[13–17] and relying on rigid printed circuit boards fabricated by conventional methods (i.e., photolithography, spin coating, and high-vacuum deposition) for the active components[15,18–20]. Since body-wearable devices typically comprise sensor elements and an electronics module, the additive process should be capable of efficiently printing various ink materials with a wide range of viscosities, and precise alignment of multiple layers. Last, biocompatibility studies should be conducted, considering that the printed materials could be in direct contact with skin for multiple days. Though the printed Ag and Cu are highly conductive and easily mass-produced, the metal ions that leach out from the metallic nanoparticles could cause adverse effects on human tissues due to their highly corrosive and oxidizing properties[21,22]. CNT, a nonmetal nanomaterial, is an attractive alternative for printing conductive elements, but its relatively low conductivity could be problematic toward reliable circuit operation and current delivery[23,24].

This work introduces all-printed, nanomembrane hybrid electronics (referred as "p-NHE") whose fabrication strategy is established by comprehensive studies on nanomaterial preparation, material processing, and printing optimization. The additive nanomanufacturing process ensures high precision alignment in multilayer printing, while the thin and flexible structures allow the printed electronics to be integrate and deform naturally with elastomers. The high-aspect ratio functionalized conductive graphene (FCG), conserving the intrinsic electrical and morphological properties, promotes cell biocompatibility and resists metal-oxidation by preventing oxygen exposure. The printed, flexible circuit allows functional chip components to be soldered to the FCG/Ag membrane, and resulting in enhanced structural reliability. To fully illustrate the feasibility of the all-printed electromyogram (EMG) devices in advancing wearable healthcare and health monitoring, we implement multiple human-machine interfaces (HMI) scenarios including hand-gesture-controlled wireless target controls, such as drones and a computer software.

Due to their compactness and low mass, multiple printed EMG devices can be applied to target muscle groups to strategically enhance the detection accuracy in complex hand gestures that typically require a large number of electrodes. To demonstrate, specific muscle groups activated during flexion of each digit are identified and three printed devices are applied for synchronized transmission of the EMG data. The synchronized multi-device EMG data, analyzed with deep-learning algorithms, are capable of real-time classification of individual digit movement for wireless control of a robotic hand. Collective results illustrate how the proposed materials optimization, device integration, and EMG-based HMIs will change the way printed electronics integrated with soft materials are utilized in advancing human performance and healthcare.

## Results

**Design and manufacturing of a p-NHE**. Figure 1 shows the overview of a new class of printing technologies and multiple nanomaterials to develop a wireless, wearable p-NHE. The main advantages of the additive manufacturing research, presented in this work, appear in Table 1, which compares recently developed methods for multilayered sensors and circuit systems[3–5,15,25–30]. Nanomanufacturing of the p-NHE (Fig. 1a) uses Ag as the conductive circuit traces, FCG as the oxidation barrier for Ag as well as sensing electrodes, and polyimide (PI) as the insulating and structural support layers. The aerosol-jet-based printing (AJP) method uses two atomizing modes (ultrasonic and pneumatic) to allow the direct deposition of inks with a wide range of viscosity from 1 to 1000 cP without the use of pattern masks or screens. The optimized atomization and sheath gas streams require a precise ejection of atomized droplets from the nozzle to the substrate (details of the optimized AJP processes in Methods, Supplementary Fig. 1, and Supplementary Table 1). In prior work, methods for formulating selectively edge-oxidized FCG were reported[31]. Carboxylic and hydroxyl groups of the FCG (Fig. 1b) facilitate the dispersion of graphene molecules in aqueous solvents without the use of dispersion agents, leading to the formation of a graphene ink highly compatible with an AJP process. This method of printing high-quality FCG offers simple yet high-resolution patterning when compared with existing strategies using graphene oxide or epitaxial growth[32–35]. In addition, by tuning the ink concentrations with solvent solutions, all inks are optimized to match the printable viscosity (Fig. 1b), thereby achieving the layer-by-layer structure required for wireless electronics. A series of printing processes are performed for two key elements of the device, including FCG electrodes for measuring electrophysiological signals (Fig. 1c) and the wireless circuit for communication with an external mobile device (Fig. 1d). As a printed output, the conductive electrodes consist of 10.5-μm-thick PI and 0.8-μm-thick FCG layers on a glass substrate, which is coated with a sacrificial polymethyl methacrylate (PMMA) layer (Fig. 1e and Supplementary Fig. 2). For the nanomembrane structured circuit, multiple layers are printed, including 0.5-μm-thick 1st conductive Ag, 2.0-μm-thick middle PI, 2.0-μm-thick 2nd Ag, 0.1-μm-thick FCG, and 1.3-μm-thick final PI (Fig. 1f and Supplementary Fig. 3). In the circuit fabrication, PI is printed to insulate the 1st Ag layer except for the circular contact spots (diameter: 50 μm) to effectively create the VIAs for the electrical connection between the 1st and 2nd Ag layers. Upon completion of the printing, the device integrates functional chip components via soldering and then the finalized device is encapsulated by a low-modulus silicone elastomer with a Young's modulus of 8.5 kPa (Supplementary Fig. 4). A more detailed description of the circuit appears in Supplementary Fig. 5 and Supplementary Table 2. The fully integrated sensors and electronics are

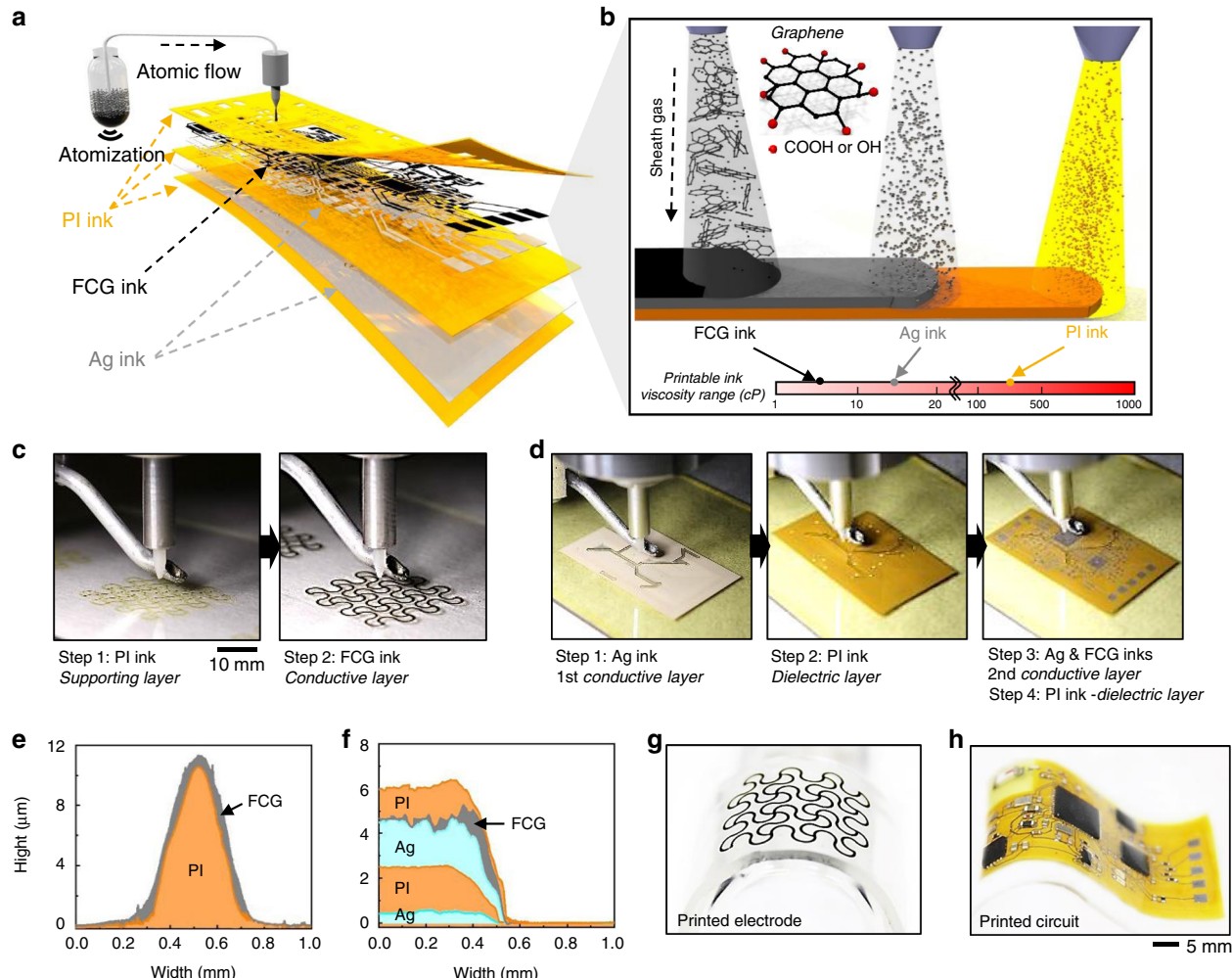

**Fig. 1 Design, architecture, and nanomanufacturing of a p-NHE. a, b** Schematic illustrations of printing of aerosol nanoparticles from PI, Ag, and graphene inks to construct a multilayered structure. The atomized droplets in a $N_2$ gas stream are directed to the print head to draw desired patterns on a substrate (**a**). The aerosol stream is controlled and delivered by a sheath gas to the substrate; three types of inks with different viscosity values are sequentially printed in this work (**b**). Small inset shows a chemical structure of graphene which is partially oxidized with carboxylic and hydroxyl groups. **c** Sequential photos of printing of PI and graphene ink to fabricate a stretchable nanomembrane electrode. In Step 1, PI ink is printed on a PMMA/glass substrate as a supporting layer. In Step 2, graphene ink is printed on top of the PI as a conductive layer for an electrode. **d** Still images showing a printing process of a multilayered biopotential recording circuit. In Step 1, Ag ink is deposited on a PI/PMMA/glass substrate as a circuit ground plane. In Step 2, PI ink is deposited as a dielectric layer except where the VIAs are located. In Step 3, Ag and FCG inks are printed as a metal interconnect layer that is connected to the ground plane. In Step 4, additional PI ink is printed to completely enclose the entire circuit except contact pads to integrate functional chip components. **e, f** Measured cross-sectional profiles of a printed stretchable electrode (**e**) and a printed multilayered circuit (**f**). **g, h** Close-up photos of the printed electrode (**g**) and circuit (**h**) on elastomeric membranes in **e** and **f**.

exceptionally light (<5 g) and thin (<2 mm) compared with conventional electronic systems, allowing conformal, intimate lamination on the skin solely by the adhesiveness of the elastomer, while being mechanically compliant for various flexible applications (Fig. 1g, h). The integrated p-NHE is powered by a miniaturized lithium-ion polymer battery (40 mAh capacity; DTP301120, Shenzhen Data Power Technology). The battery's two terminals and the circuit's power pads are soldered with small neodymium magnets for a guided battery connection. The details for battery connection and power efficiency are described in Supplementary Fig. 6.

**Fabrication and characterization of nanomembrane FCG sensors.** For health monitoring and human-machine interfaces (HMI), it is critical to employ a highly conductive and compliant electrode that can seamlessly interact with the human skin[36–39]. In this study, we developed a nanomanufacturing process to design a stretchable, nanomembrane FCG sensor. We used atomic force microscopy (AFM) to characterize the prepared FCG ink. AFM imaging and measured size histograms in Fig. 2a, b reveal that the prepared FCG has lateral dimensions ranging between 1 and 5 μm and an estimated thickness of 3.1 nm, confirming the high-aspect-ratio structure (Supplementary Fig. 7a, b). The thickness distribution of 100 samples of FCG sheets from the AFM measurement is shown in Fig. 2b. In addition, we used the transmission electron microscopy (TEM) to study the nanostructure of FCG. The captured image in Fig. 2c shows the FCG structure with two micrometer-wide walls, which is in agreement with the AFM results. Furthermore, the high resolution TEM image in Fig. 2d shows about 1.5-nm-thick FCG structure at the edge of a bilayer graphene. The electron density patterns at the basal plane of FCG reveal the sixfold symmetry of the hexagonal structure, confirming that the inner basal plane of FCG is well preserved (Supplementary

**Table 1 Comparison of multilayered sensor and circuit systems.**

| Reference | Fabrication method | Sensor materials | Circuit materials | Sensor type | Acquisition mode |
|---|---|---|---|---|---|
| This work | All-printed | Graphene/PI on the elastomer | PI/Graphene/Ag/PI/Ag/PI on the elastomer | EMG | Synchronized, multi-device |
| 3 | Microfabrication (sensor) | Au/Cr on the adhesive silicone | None | EMG | Single device |
| 4 | Casting (sensor) Laser ablation (circuit) | CNT and Silbione | Four layers with a Cu/PI on the elastomer | EMG | Single device |
| 5 | Microfabrication (sensor) | PI/IZO/Au/Cr/PI on the elastomer | None | Strain | Single device |
| 25 | Printed (sensor) Microfabrication (circuit) | Ag/PI on the elastomer | PI/Cu/PI/Cu/PI on the elastomer | EOG | Single device |
| 15 | Printed (sensor) | Ag/AgCl on the polyester | Rigid circuit board | ECG and lactate | Single device |
| 26 | Microfabrication (sensor and circuit) | PI/Au/Cr/PI on the elastomer | Cu/PI/Cu on the adhesive | Temperature | Single device |
| 27 | Laser ablation (sensor) | PI/Cu/transducer/Cu/PI on the elastomer | None | Blood pressure | Single device |
| 28 | Laser ablation (sensor) | PI/Cu/transducer/Cu/PI on the elastomer | None | Ultrasonic probe | Single device |
| 29 | Microfabrication (sensor and circuit) | PI/Cu/PI/Cu/PI on the elastomer | PI/Cu/PI/Cu | Blood flow | Single device |
| 30 | Microfabrication (sensor) | PI/Cu/PI on the elastomer | None | Thermal | Single device |

Fig. 7c). Such a high-aspect ratio FCG in an ink droplet leads to a plane-to-plane stacking structure after printing (Fig. 2e). Cross-sectional scanning electron microscope (SEM) images in Fig. 2f, g validate that the printed FCG is clearly well-stacked and integrated on the printed PI and elastomeric membrane.

To validate the potential of the printed FCG as a wearable electrophysiological sensor, we conducted a series of experimental studies. As a feasibility test, we designed skin-wearable FCG electrodes and performed an optimization study of the electrode design with different sizes (8–16 mm in diameter; Supplementary Fig. 8). With the chosen 16-mm-diameter electrode, EMG signals were measured, and the signal quality was compared with the gold-standard gel-based electrode and typical metal electrodes (Au and Ag). The summarized results in Fig. 2h–j capture the outstanding performance of the FCG electrode over other metals. The signal-to-noise ratio (SNR) in Fig. 2j clearly shows that the multilayered, 3D graphene structure has enhanced contact with the skin compared with the 2D sheets of printed Au and Ag layers (Supplementary Fig. 9 and Supplementary Table 3). Even though the FCG electrode has a slightly lower SNR than that of the gel electrode, the difference is negligible since the difference is within in the error range. In addition, the conventional gel electrode has disadvantages of skin irritation/allergic reactions and fluctuation of the skin-electrode contact impedance due to drying gels over time[2,39]. In addition, structural reliability and mechanical integrity analyses were performed on the printed FCG electrode in stretching and bending modalities. Computational study with finite element analysis (FEA) was used to design a highly bendable (up to 180° with 1.5 mm in radius) and stretchable (up to 60%) electrode (Supplementary Figs. 10 and 11, and Supplementary Table 4a). Corresponding experimental study validated the estimated mechanical safety of the FCG electrode upon excessive bending and stretching, which was quantified by measuring the change in the electrical resistance. The regression plots in Supplementary Figs. 10e and 11e capture that the printed electrode has performs consistently during the cyclic testing of 100 repetitions as measured in the coefficient of determination of $R^2 = 0.98$ and 0.99, respectively.

The open-mesh structured FCG electrode shows highly conformal lamination on the human skin (Fig. 2k), maintaining contact quality during extension and compression. Toward validating biocompatibility with the skin, a cytotoxicity study of the fabricated FCG electrode was conducted with human keratinocyte cells. The results of a week-long cell culture on multiple samples are shown in Fig. 2l–n. After 7 days of the human cell culture, the number of live cells on five different substrates is determined via fluorescence intensity. As shown in Fig. 2l, the graphene sensor shows comparable cell viability to the control sample (polystyrene cell culture dish). Similarly, other materials such as an elastomeric membrane and printed Au also shows a healthy cell culture environment based on the cell proliferation rates for absorbance and fluorescence (Fig. 2m–n and Supplementary Fig. 12). However, printed Ag demonstrates the unwanted cytotoxic condition to the human cells. According to prior studies on cytotoxicity[21,40], Ag nanomaterials are easily oxidized in biological and environmental media, resulting in the release of ions. The released metal ions interact with membrane proteins, damaging proteins and nucleic acid, and finally inhibiting cell activation. Overall, these studies guided our decision to utilize the FCG as an oxidation barrier for a circuit manufacturing and as a sensor for the direct contact to the skin.

**Characterization of a multilayered electronic circuit**. The all-printed flexible circuit is composed of multiple layers of printed nanomembranes and onboard functional chip components

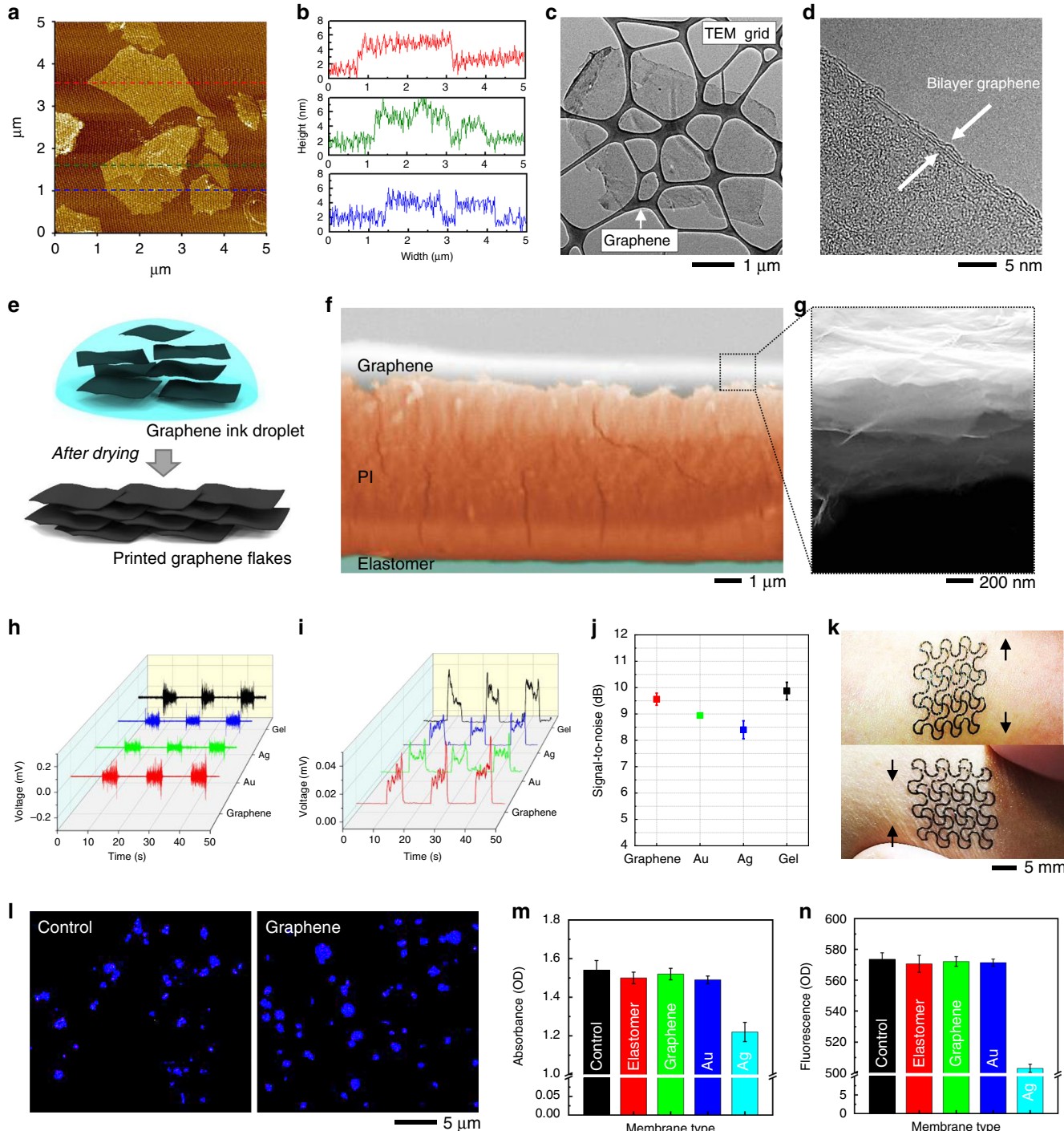

**Fig. 2 Fabrication and characterization of a FCG and its application as a sensor. a** AFM image (top view) of a FCG that was initially dispersed in aqueous ink on a mica substrate. **b** Three height profiles of FCG flakes, corresponding to three dotted lines in **a**. **c, d** TEM images of a FCG with a low-resolution view (**c**) and high-resolution view (**d**). **e** Schematic illustration representing the stacking process of printed FCG before and after drying of an ink droplet. **f** Colorized, cross-sectional SEM image, showing a multilayered sensor structure, including graphene, PI, and elastomeric substrate. **g** Enlarged view of the SEM image in **f**, showing the stacked FCG layers with total thickness of 800 nm. **h** Recorded EMG signals on forearm that compare the signal amplitudes with four electrodes, made of printed FCG (red), Au (green), Ag (blue), and commercial metal/gel (black). **i** Root-mean-squared data of EMG signals from (**h**). **j** Averaged signal-to-noise ratio of EMG data from a printed FCG (red), Au (green), Ag (blue), and commercial gel electrode (black): error bars show standard deviation (*n* = 3). **k** Optical images showing a conformal lamination and stretchability of the FCG electrode on the skin: longitudinal stretching (top) and compression (bottom) on forearm. **l** Optical microscopic images of cultured keratinocyte cells after exposure to original DMEM (left; control sample) and DMEM with graphene (right) after 7 days of culture. **m, n** Comparison of cell absorbance (**m**) and fluorescence intensity (**n**) of cultured keratinocyte cells on five types of membranes, including control, elastomer, graphene, Au, and Ag. Overall, these data capture the cell friendly environments of a graphene electrode on an elastomeric membrane.

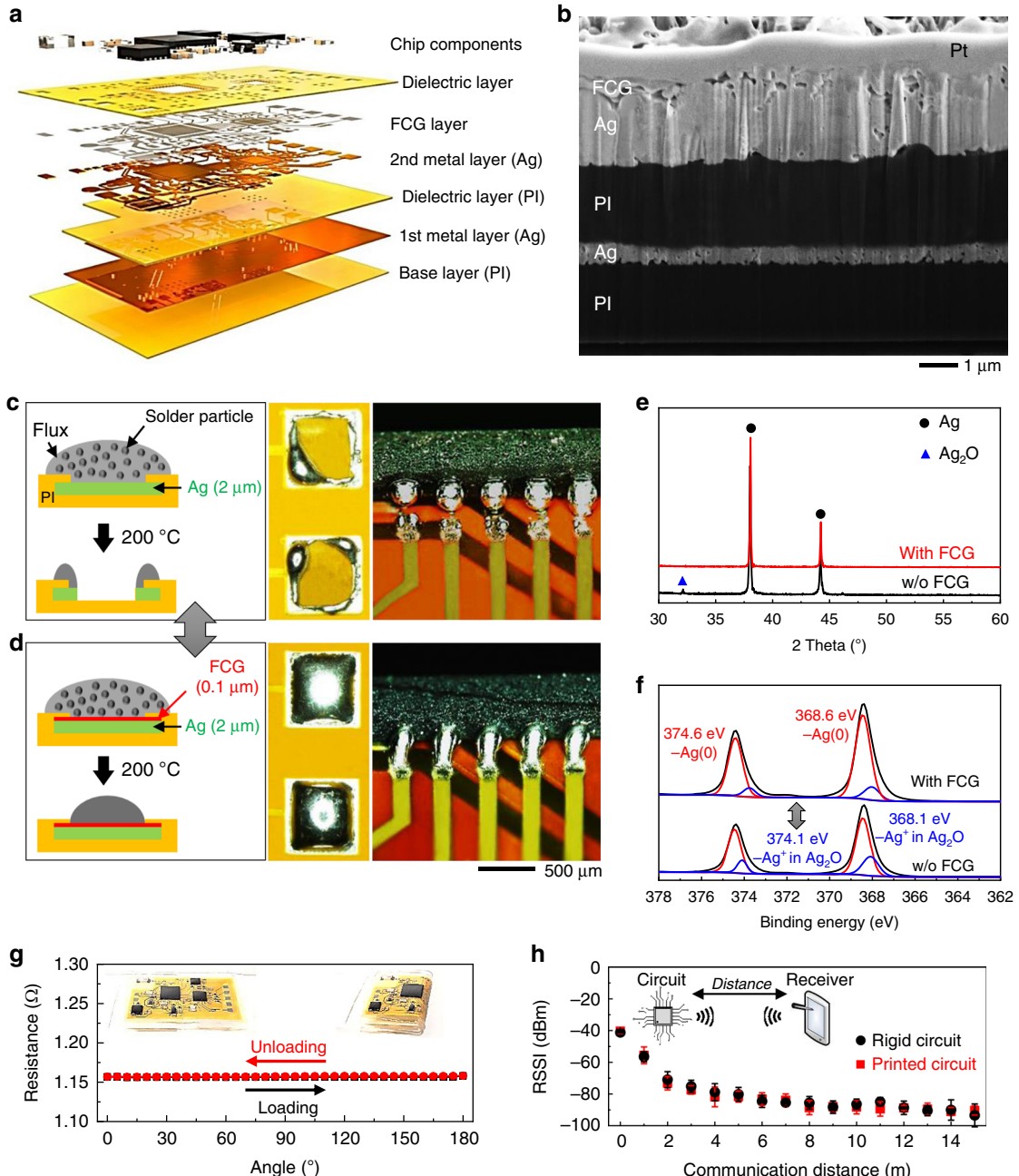

**Fig. 3 Printing, integration, and characterization of a multilayered electronic circuit. a** Schematic illustration that captures the multilayered, printed, flexible, wireless electronics. **b** FIB-assisted SEM image showing the cross-section of a printed multilayered circuit that follows the illustration in **a**; Pt is deposited on top of the circuit to protect the circuit structure during preparation for a cross-section imaging. **c**, **d** Comparison of solderability between two materials. Schematic illustration (left side on **c**, **d**) shows two different circuits with Ag layer only (**c**) and FCG (thickness: 0.1 μm) on top of Ag layer (**d**), which captures the capability of FCG to block the consumption of the Ag layer due to the solder flux. Corresponding photos (middle and right) clearly show the difference between two cases; without FCG, the Ag-based contact pad is removed during the reflow step. **e**, **f** Characterization results with XRD (**e**) and XPS (**f**) of sintered Ag with and without the printed FCG. XRD data (**e**) capture the anti-oxidation effect of printed FCG, while XPS data (**f**) show the FCG/Ag membrane is only diffracted at the peaks of 38.0°, 44.1°, and 54.8°, indicating the pure Ag structure without oxidized Ag. **g** Change of electrical resistance of a highly flexible printed circuit upon 180° bending (up to 100 cycles), showing negligible change in resistance. Inset photos show the tested circuit wrapped around a curved glass edge up to 180°. **h** Simultaneous comparison of RSSI response between a commercial rigid circuit and printed circuit, showing a comparable performance of the printed circuit with the communication distance up to 20 m based on Bluetooth.

(Fig. 3a). The cross-sectional SEM image in Fig. 3b captures the uniformity of the layer-by-layer structure with great packing density around one of the conductive pads to be connected with chips. This image also shows the capability of the AJP method to allow reliable deposition of multiple materials sequentially. In the circuit, the conductive Ag traces were photonically sintered via a

processing condition optimization. Sequential SEM images in Supplementary Fig. 13 show the structural change from as-printed Ag nanoparticles to a densely-packed Ag membrane for a conductive film.

For the development of a wireless electrophysiological monitor, a set of miniaturized functional chips were utilized, including

microcontroller/Bluetooth, amplifier, antenna, voltage regulator and more (details in Supplementary Fig. 5). A known issue regarding Ag conductors is surface oxidation since the conventional reflow soldering process uses a high temperature above 140 °C, which limits the direct integration of chips to the printed circuit (Fig. 3c). The oxidized metal reacts with the flux of the solder paste to dissolve the pads (Fig. 3c), which is clearly visible in the microscopic images (Fig. 3c). On the other hand, the FCG/ Ag pads (Fig. 3d) achieve great solderability with solder paste as the printed nanomembrane FCG functions as an oxidation barrier. X-ray diffraction (XRD) study in Fig. 3e confirms this critical role of the printed FCG. The sample without the FCG reveals a peak at 32.7° that is related to the cubic structure of $Ag_2O$ (ICDD card No. 00-41-1104). By comparison, the FCG/Ag membrane is only diffracted at the peaks of 38.0°, 44.1°, and 54.8° (ICDD card No. 00-004-0783), indicating the pure Ag structure. X-ray photoelectron spectroscopy (XPS) is also used to analyze the compositional structure of the printed Ag with and without the FCG (Fig. 3f). Curve fitting of Ag (top graph; with FCG) shows two strong peaks at 368.6 and 374.6 eV, which are consistent with Ag(0) metal, while the bottom graph without FCG shows two weak peaks at 368.1 and 374.1 eV, which correspond to $Ag^+$ ions in oxidized Ag. In addition, the relative elemental composition showing reduced oxygen atomic concentration reveals the oxidation stability of the FCG/Ag traces (Supplementary Table 5). High-resolution XPS of Ag ranging from 366 to 371 eV indicates that the ratio of $Ag^+/Ag(0)$ is reduced from 0.34 of Ag without FCG to 0.16 of Ag with FCG, suggesting that FCG prevents the oxidation of printed Ag (Supplementary Fig. 14).

Additional studies in computational and experimental mechanics in Fig. 3g and Supplementary Fig. 15 demonstrate the mechanical reliability of the printed flexible circuit upon excessive 180° bending cycles. FEA of circuit interconnects suggests a reliable circuit design with the maximum principal strain below 1% (Supplementary Fig. 15a), which is validated in a cyclic bending test (Supplementary Fig. 15b). Figure 3g shows negligible change in electrical resistance over 100 cycles. In addition, wireless transmitted acceleration data during the bending test (Supplementary Fig. 15c–e and Supplementary Table 4b) validate the mechanical safety of the circuit that would experience time-dynamic skin deformations. Similarly, the wireless Bluetooth performance of the printed circuit is compared with the commercial rigid circuit board (Fig. 3h). As a result, the received signal strength indicator (RSSI) measured from the printed circuit displays consistent signals up to 15 m, which is comparable to the rigid circuit (Fig. 3h and Supplementary Table 6). For the reliability study of the integrated p-NHE, we conducted EMG recording and SNR comparison before and after 100 bending cycles (Supplementary Fig. 16 and Supplementary Table 4b), which shows a negligible change of the EMG quality. In addition, the fabrication yield was calculated based on each process, including the printing, soldering, and integration (Supplementary Table 7). The optimized printing and soldering processes provide a high yield over 90%, while the final integration of sensor and circuit reduces the yield. Once an automated device assembly is implemented, then the overall yield will be significantly improved. Collectively, the wireless p-NHE using the FCG nanomembrane demonstrates exceptional solderability of chip components, reliable mechanical flexibility, and stable, long-range wireless communication.

**Demonstration of the wearable p-NHE for multi-class HMIs.** This study summarizes the implementation of the wearable p-NHE for multi-class HMIs, which capture the potential of the developed nanosystem for smart rehabilitation, advanced therapeutics, and other machine-integrated applications. The exceptionally small form factor and lightweight of the p-NHE allows for a comfortable, seamless lamination to skin areas. An example in Fig. 4a shows mounting of three skin-resembled, p-NHE on the forearm that wirelessly monitor non-invasive EMG via a long-range, high-throughput Bluetooth. The portable and wearable system can utilize measured EMG data for a real-time, wireless, and continuous control of external devices in daily life. To acquire all motions of five fingers and wrist flexion, prior works[41–43] required arrays of over 10 EMG sensors to cover the entire forearm, mainly due to the poor spatial resolution and motion artifacts from wires and gel-electrode slippage. In this study, we investigated the major muscle groups that connect all fingers to the forearm (see the illustration in Fig. 4a), resulting in an EMG heat map that shows the signal strength caused by each finger motion (Fig. 4b). Details of the experimental method for the heat map generation are described in Methods and Supplementary Fig. 17. Using this heat map, average root-mean-square (RMS) values are calculated across the entire forearm to determine optimal device mounting locations.

As a first example, a single p-NHE is mounted on the muscle, palmaris longus, which shows one of the strongest signals in Fig. 4b. To demonstrate a multi-class HMI (Supplementary Fig. 18) using wearable devices and machine-learning algorithms, a subject wears the p-NHE to generate several motions, including open hand, closed hand, flexion of index finger, and wrist flexion. The corresponding raw EMG data are shown in Fig. 4c. For safe, real-time control of external systems, accurate classification of acquired data is critical. We studied two types of machine-learning algorithms including convolutional neural networks (CNN) and k-nearest neighbors (KNN; additional details appear in Methods). A series of flowcharts in Supplementary Fig. 19 summarizes how the p-NHE and algorithms are used to enable multi-class HMIs. The single device-enabled EMG (Fig. 4c) and z-axis acceleration (Supplementary Fig. 20) are utilized to control three external targets, including a quadcopter drone, RC car, and presentation software (Microsoft PowerPoint; Fig. 4d). Based on the 2-layer CNN (Supplementary Table 8), this HMI system shows a high classification accuracy over 99% with six classes over 10-trials for real-time control (Supplementary Figs. 21 and 22 and real-time demonstration of machine controls in Supplementary Movies 1–3). Details of the hand gestures and intended commands for three target machines are summarized in Supplementary Table 9.

In addition, for a higher number of control classes, we utilized three p-NHEs and mounted them on three physiologically relevant muscles: palmaris longus, brachioradialis, and flexor carpi ulnaris (Fig. 4a). These locations on the forearm have the strongest EMG signals compared with other areas, enabling classification of individual digit controls and multiple hand gestures (open hand, closed hand, and flexion of five different fingers). A 3D, three-channel RMS plot from three devices (Fig. 4e) shows seven distinctive clusters, generated by motions of individual fingers and hand gestures over repeated trials. A set of three p-NHEs, distributed over three different muscle groups, clearly captures different RMS values per each motion, as summarized in Supplementary Table 10. The resultant confusion matrix in Fig. 4f, generated by the developed classification algorithm, shows the overall accuracy of 98.6% for seven classes over ten different trials from multiple subjects. To demonstrate the capability of a multi-class HMI, we manufactured a robotic hand via 3D printing and materials assembly. The classified EMG signals from three wearable devices allow wireless, real-time control of the robotic hand in Fig. 4g (details in Supplementary Fig. 23 and Supplementary Movie 4). Overall, the soft, skin-

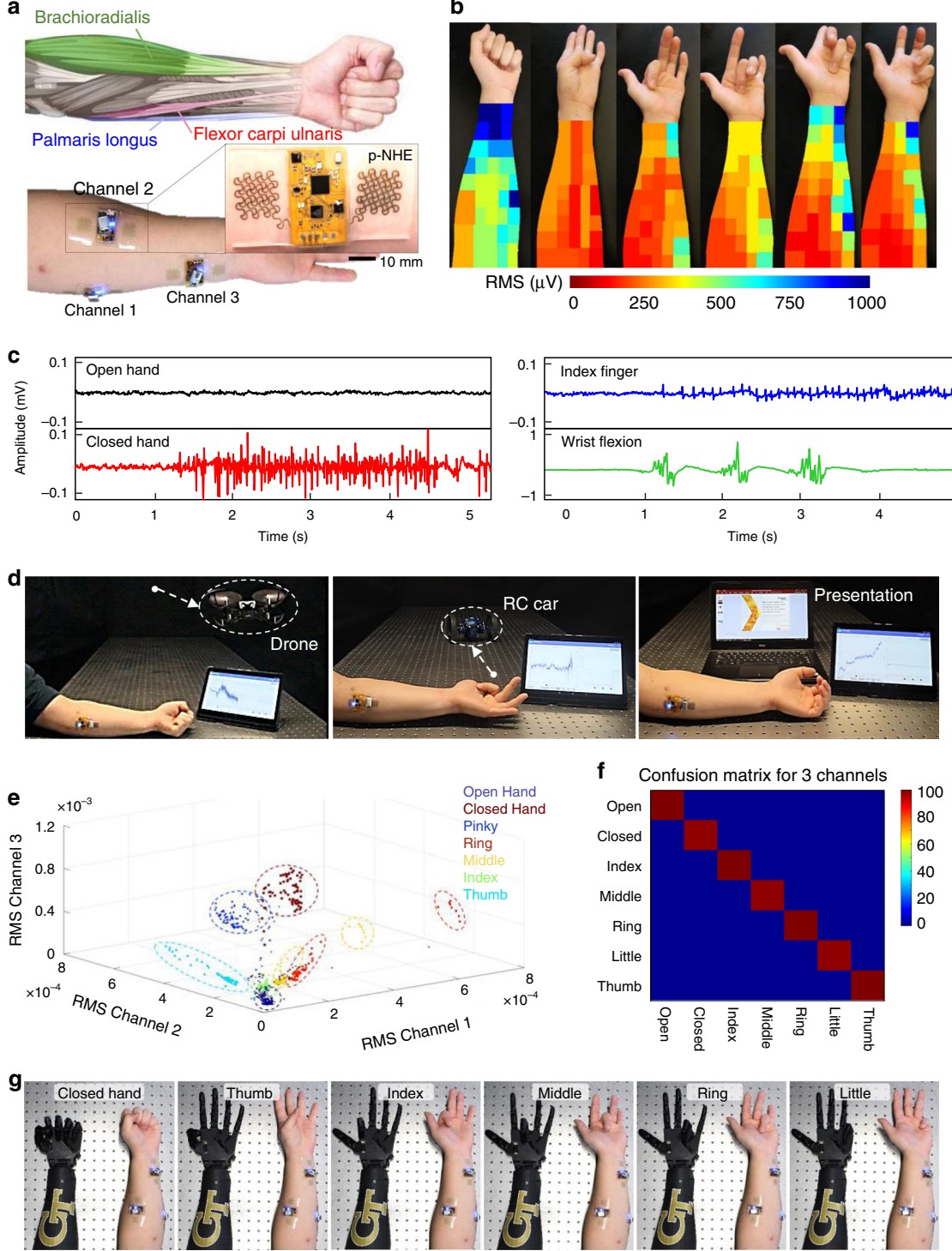

**Fig. 4 Demonstration of a wearable p-NHE for wireless, multi-class human-machine interfaces. a** Schematic illustration (top) showing target muscles on forearm to recognize multiple gestures and photos (bottom) capturing three p-NHE positioned on targeted muscles, including palmaris longus, brachioradialis, and flexor carpi ulnaris. Enlarged image shows one of those systems with a circuit and electrodes. **b** EMG mapping data showing RMS signals from multiple channels that cover the entire forearm; six different heat maps are obtained according to six gestures, including hand closed, thumb, index, middle, ring, and little finger motion (from left photo to right photo), for determining the ideal channel locations. **c** Representative EMG signals from four gestures, including open hand, closed hand (left graphs) and index finger flexion and wrist flexion (right graphs). **d** Demonstration of EMG-enabled human-machine interfaces with a wireless, wearable p-NHE to precisely control a flying drone (left), RC car (middle), and presentation software (right). In this test, a single-channel device is mounted on the muscle, palmaris longus (shown in **a**). **e** 3D plot of three-channel, EMG RMS signals for clear differentiation of seven different gestures as seven groups. **f** Summarized result of real-time, confusion matrix from ten trials, showing 98.6% accuracy across seven classes with three-channel EMG recording. **g** Demonstration of a three-channel EMG recording and corresponding control of a robotic hand, showing examples of six motions of a subject and following motions of the robotic hand.

friendly, p-NHE shows great potential for various portable HMI applications including control of a humanoid robot, drone, prosthetic hand, display interface, electronic wheelchair, and more[39].

## Discussion

Collectively, this paper reports an additive nanomanufacturing of functional nanomaterials and polymers that enables a wireless, multilayered, seamlessly interconnected p-NHE. This work illustrates the first demonstration of an all-printed, nanomembrane electronics using multiple nanomaterials to construct high-performance, wearable sensors and wireless circuits. Biocompatible, high-aspect ratio FCG nanomaterial offers excellent conformal lamination on human skin for high-fidelity recording of EMG, while providing reliable solderability by preventing Ag oxidation. A set of comprehensive experimental and computational studies validates mechanical stretchability of the sensor and flexibility of the circuit to endure time-dynamic, multimodal strains from the wearable applications. Machine-learning integration with the wearable p-NHE demonstrates multiple HMI use cases, including optimal selection of three sensor channels from a larger electrode cluster. These methods show a successful detection of all finger motions with an accuracy of about 99% for seven classes. Future studies will focus on clinical applications of the wearable p-NHE for a biofeedback-enabled prosthetic development and enhanced rehabilitation training.

## Methods

**Ink preparation**. Preparation of the FCG, Ag, and PI inks was conducted using the process we reported in our previous research[31,44]. For synthesis of FCG ink, a direct voltage of 10 V was applied between the graphite (Alfa Aesar) and Pt foil in an electrolyte solution of ammonium sulfate ($(NH_4)_2SO_4$, Sigma-Aldrich). After exfoliation, as-synthesized graphene was purified using deionized water (DI water) and further filtered under vacuum to remove the residuals. The filtered wet powder of graphene was dispersed in DI water and controlled to the concentration of 15%. An Ag nanoparticle ink (Ag40XL, UT Dots) was mixed with xylene (m-Xylene, Sigma-Aldrich) to make the Ag concentration of 20%. The PI ink is composed of a mixture of a precursor (PI-2545, DuPont) and solvent (1-methyl-2-pyrrolidinone; NMP, Sigma-Aldrich) in a 4:1 ratio.

**Electrode printing**. The all-additive process was performed via the AJP method (Aerosol Jet 200, Optomec), and the optimized conditions are shown in Supplementary Table 1. A sacrificial layer of PMMA (950 PMMA, Kayaku Advanced Materials) was spin-coated on a glass slide at 1000 rpm for 30 s and baked at 200 °C for 2 min. A PI ink is atomized in the pneumatic atomizer and deposited using a 300-μm-diameter nozzle. The printed PI was cured at 250 °C for 1 h. To deposit FCG ink onto the PI layer, the print head assembled with a 200-μm-diameter nozzle was precisely aligned. The FCG ink was printed via the ultrasonic mode and a 300-μm-diameter nozzle. The resulting traces were thermally cured at 100 °C for 1 h to dry the DI water in the FCG ink. The elastomer substrate was prepared by mixing 5 g of 1:1 Ecoflex00-30 (Smooth-On) and 5 g of 1:1 Ecoflex Gel (Smooth-On). Ten grams of the mixture was poured into a polystyrene dish (FB0875714, Fisher Scientific) and cured at room temperature overnight. The cured elastomer was removed from the dish for thin-film integration. To transfer the printed electrodes onto the prepared elastomer substrate, the PMMA of the printed electrodes was dissolved in an acetone bath overnight. The patterns were peeled off with water-soluble tape. Finally, the printed electrodes were transferred to the prepared 500-μm-thick Ecoflex. The printing process for the electrodes is illustrated in Supplementary Fig. 1a.

**Circuit printing**. A sacrificial layer of PMMA was coated on a glass slide at 1000 rpm for 30 s. The spin-coated PMMA/glass substrate was baked at 200 °C for 2 min. PI ink was spin-coated on a PMMA/glass substrate. Ag ink was printed onto the aligned PI layer using the ultrasonic mode and a 200-μm-diameter nozzle. The Ag ink was sintered with an intense pulsed light (IPL) equipment (S-2200, XENON Corp.). An optimal sintering condition was determined to be 2 kV/2 ms/5 times, for power and the number of pulses, respectively, that yielded the bright appearances in the resulting Ag. The PI ink for dielectric layer was printed on the 1st Ag layer via the pneumatic atomizer and the 300-μm-diameter nozzle. The printed PI was cured at 250 °C for 1 h. The 2nd Ag layer was printed and photonic-sintered with the identical parameters. To prevent the oxidation of Ag layers, the solderable FCG ink was printed and dried at 100 °C for 1 h. The final PI layer was deposited for the encapsulation and thermally cured at 250 °C for 1 h. After dissolving the

sacrificial PMMA layer in an acetone bath, the printed circuit was transferred to the 500-μm-thick Ecoflex. The printing process for the circuit is illustrated in Supplementary Fig. 1b.

**Integration with electronic components**. Solder paste (alloy of Sn/Bi/Ag (42%/57.6%/0.4%), ChipQuik Inc.) was screen-printed with a stainless-steel stencil on the top surface of the circuit. The chip components, including the BLE, ADC, voltage regulator, resistors, inductors, and capacitors, were mounted and reflowed by applying heat according to the temperature recommended by the solder paste manufacturer. The firmware of the Bluetooth-microcontroller was updated on the soldered circuit. Connecting the electrodes and the circuit with a connector completed device preparation. Chip mount information is listed in Supplementary Fig. 5 and Table 2.

**Characterization of materials**. A profilometer (Dektak 150, Veeco) was used for measuring each printed layer. Microscopy images were taken by scanning electron microscopy combined with a focused ion beam (FIB-SEM; Nova Nanolab 200, FEI), TEM (JEM-2100F, JEOL), AFM (XE-100, Park System), and optical microscopy (VHX-600, Keyence). The crystallographic and elemental structures were analyzed via XRD (X'Pert PRO Alpha-1, Malvern Panalytical) and XPS (K-Alpha XPS, Thermo Fisher). All mechanical tests were performed using a digital force gauge (M5-5, Mark-10) fixed on a motorized test stand (ESM303, Mark-10).

**Study of finite element analysis (FEA)**. FEA estimated the mechanical behavior of the printed device by using ABAQUS software (Dassault Systemes Simulia Corporation, Johnston, RI). The modeling study focused on the mechanical reliability of the multilayered device upon repetitive bending and stretching to simulate the mechanical environment of a forearm. The simulation used the following material properties[45]: Young's modulus ($E$) and Poisson's ratio ($v$): $E_{PI}$ = 2.5 GPa and $v_{PI}$ = 0.34 for PI; $E_{Ag}$ = 40 GPa and $v_{Ag}$ = 0.37 for Ag.

**Study of biocompatibility via cell viability measurement**. To ensure biocompatibility of the printed electrodes, cell viability was analyzed under multiple conditions. Tests used human primary keratinocyte cells cultured in four different conditions—control; FCG; Ag; and Au—in an incubator at 37 °C with 5% CO₂. In the incubator, the material samples were placed in a 24-well plate, and 5000 keratinocytes/cm² were seeded. After 7 days in the incubator, keratinocyte cells were washed with phosphate-buffered saline (Fisher Chemical) and dyed with 0.1 ml of calcein blue AM (Thermo Fisher) in 0.9 ml of the cultured medium. Keratinocytes and the reagent were additionally stored in the incubator of 37 °C for 10 min. The supernatant was then aliquoted in a 96-well plate for further biocompatibility. To quantify and compare the cell viability of four typed samples, a multi-mode microplate reader (SpectraMax, M3) was used.

**Generation of EMG heat maps**. Identification of specific muscle groups activated during each of the seven hand gestures was carried out by covering the left forearm with multiple 1′ × 1′ hydrogel electrodes (MVAP-II, MVAP Medical Supplies). For EMG acquisition, a commercial system (BioRadio, Great Lakes Neurotechnologies) was used with a three-electrode setup. Measurement and reference electrodes were selected as an adjacent pair on the left forearm, whereas the ground electrode was attached to the bony part of the right wrist. For each measurement location, a gesture was performed three times and the signals were bandpass-filtered from 30 to 150 Hz with a notch filter at 60 Hz. The filtered signals were converted to RMS to determine both the signal and the noise amplitudes, where the baseline noise and signal peak were defined as the data collected during the resting period and the maximum signal amplitude during each gesture, respectively.

**In vivo experiment with human subjects**. The study involved volunteers aged between 18 and 40 and was conducted by following the approved IRB protocol (# H17212) at Georgia Institute of Technology. Prior to the study, all subjects agreed with the study procedures and provided signed consent forms.

**HMI-single device classification**. Data from the EMG device were transmitted using Bluetooth Low-Energy protocols using a chip antenna soldered onto the circuit. These data were received with an Android tablet and processed in a custom Android application. A flow chart demonstrating the data flow is provided in Supplementary Fig. 17a. For a single device, a data buffer for classification was updated every time a new packet is received. The final 0.512 s of data (128 data points) was processed and classified for every 10 packets received (60 data points or 0.240 s). Due to the fast classification rate, and the nature of EMG control, a safety buffer was used to buffer the classification output by collecting the last five classification outputs. If all five outputs match, then the command was sent. This created a cumulative delay of 1.2 s for every new command. However, the commands were halted if there was any confusion in the buffer. This made the operation much safer as it reduced translation of both classification and human error into target commands. Therefore, convolutional neural networks were used to recognize subtle firing rate patterns while also taking into account the amplitude. For preprocessing, the data were only high-pass-filtered using a 3rd-order

Butterworth filter at a cutoff frequency of 1 Hz, and linearly scaled up so the highest fluctuations did not exceed ±1 V. A simple 2-layer CNN model (Supplementary Table 8) was trained using 33.3 min of training data from the five subjects. These included ten 40-s recordings of each task repeated 10 times for 4 s for each task. These data were split into 25 min of training data and 8.3 min of test data using a fourfold cross validation method. The resulting mean accuracy was 99.0 ± 1.7%. As 4 commands were not sufficient for all HMI tasks, additional simple motion controls were implemented using the onboard accelerometer for the quadcopter drone and RC car (Supplementary Fig. 17b). Here, rotation of the arm was tracked using the $z$-axis acceleration, with programmed cutoffs, as shown in Supplementary Fig. 18. For the drone, this was used to raise or lower the flying altitude. The classification model was tested using the same protocol as the training data by a 5th subject, whose data were not included in the training data. As a result, a mean real-time accuracy of 99.2 ± 1.0% was achieved over 10 trials (Supplementary Fig. 20), which compared favorably with the offline testing.

**HMI-three device classification**. The configuration for controlling the robotic hand required a greater number of channels to decipher a greater number of hand gestures. In this case, seven gestures were monitored using three EMG channels in a bipolar arrangement, each running on a separate flexible device, as explained in the main text. The data from the three devices was synchronized on a per-session basis and required that the dataset was retrained for each session. This avoided the need to connect to the devices in a specific sequence in order to synchronize the retrieval of packets over multiple sessions. Therefore, the subjects underwent a short training session before the interface was used. The protocol cycled through each instruction with a rest (open hand) step in-between to allow the subject to relax. Each command was performed for 5 s and repeated twice, resulting in a total training time of 140 s. As the required dataset for classification was quite small, and the data were easily separable (Fig. 4f and Supplementary Fig. 21), a KNN approach was used to classify the data. This instance-based approach was favorable due to the ease in separating the data, and the small size of the training set. As with the single device EMG interfaces, a control buffer was used to ensure incorrect commands were not sent accidentally (Supplementary Fig. 17c). Real-time session results based on four sessions are show in Supplementary Fig. 20.

## Data availability

The data generated and/or analyzed, and computer codes developed during this study are available from the corresponding author upon request.

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

## Acknowledgements

We acknowledge the support by the Georgia Research Alliance based in Atlanta, Georgia. This work was partially supported by the National Institutes of Health under award number (NIH R21AG064309). The content is solely the responsibility of the authors and does not necessarily represent the official views of the NIH. Device preparation was partially supported by the Nano-Material Technology Development Program through the National Research Foundation of Korea (NRF) funded by the Ministry of Science, ICT, and Future Planning (2016M3A7B4900044). This work was performed in part at the Georgia Tech Institute for Electronics and Nanotechnology, a member of the National Nanotechnology Coordinated Infrastructure (NNCI), which is supported by the National Science Foundation (ECCS-1542174).

## Author contributions

Y.-T.K., Y.-S.K., and W.-H.Y. designed the experiments. Y.-T.K., Y.-S.K., and S.K. performed device integrations. Y.-T.K., Y.-S.K., H.-R.L., S.-W.P., S.-O.K., and R.H. performed experiments and analysis. S.K. performed computational modeling. Y.-T.K., J.J.C., and Y.C.J. performed cell viability experiments and analysis. Y.-T.K., Y.-S.K., and M.M. performed HMI experiments and data analysis. Y.-T.K., Y.-S.K., M.M., Y.-H.C., and W.-H.Y. wrote the paper.

## Competing interests

W.-H.Y. and Y.-T.K. are the inventors on a pending US patent application.
