## [Peer Review File · Nature Communications]

Reviewers' Comments:

Reviewer #2:

Remarks to the Author:

The paper presents a rigorous study of a printed skin mounted system for EMG pickup and wireless transfer. The work has some particular points of interest in using graphene and silver inks to create stretchable electrodes, and also to create a system which allows effective solder reflow.

The supporting biopotential measurement multilayer electronics is also printed, but what was the battery and how was it connected?

Fig.4 shows different finger gestures used to control a robotic hand, but how was it possible to isolate the ring finger from the litter finger in the final photo?

Finally, there are various minor grammatical errors or typos in the manuscript.

Reviewer #3:

Remarks to the Author:

This paper presents an additive manufacturing process to create multi-layer circuit boards for wearable devices. The overall approach is interesting and reasonably well executed. However, the authors do not contrast their approach to that from J. Rogers (Northwestern) or S. Xu (UCSD), whom have made multi-layer flexible circuit technology in the recent past, which includes the ability to mount commercial chips onto the developed substrates. The authors are strongly encouraged to explicitly compare their approach to these others, stating both the advantages and disadvantages.

One of the biggest challenges facing flexible electronics in general - even those developed with commercially-available techniques, is reliability. The authors touch on this on slightly in the manuscript by demonstrating functional operation during bending. However, the results are not convincing. The paper would be strengthened considerably if results for yield (i.e. how many of the fabricated devices had full or partial functionality), and more repeated stress tests were reported.

Lastly, the authors do not properly describe how this system is powered. More details in this regard are needed.

Reviewer #2 Remark:

The paper presents a rigorous study of a printed skin mounted system for EMG pickup and wireless transfer. The work has some particular points of interest in using graphene and silver inks to create stretchable electrodes, and also to create a system which allows effective solder reflow.

Comment #1: “The supporting biopotential measurement multilayer electronics is also printed, but what was the battery and how was it connected?”

Our response:

We thank the reviewer for pointing this out. In fact, the electronic system is powered by a lithium-ion polymer battery (40 mAh; DTP301120, Shenzhen Data Power Technology) as shown in the figure below. The miniaturized neodymium magnets facilitate an easy and secure connection of the battery, which are placed on both the circuit and the battery. When the battery is attached, it is indicated by the green LED, while the blue LED shows the connection of Bluetooth between the device and external tablet. In this figure, we added an experimental result that shows the battery (40 mAh) operation time. Depending on the device applications, data sampling rate and battery capacity can be changed to provide sufficient power.

Our modification to the manuscript:

The following description about the battery is added in Page 3 of the Manuscript:

“The integrated p-NHE is powered by a miniaturized lithium-ion polymer battery (40 mAh capacity; DTP301120, Shenzhen Data Power Technology). The battery’s two terminals and the circuit’s power pads are soldered with small neodymium magnets for a guided battery connection. The details for battery connection and power efficiency are described in Supplementary Fig. 6.”

In addition, the following figure is added in the Supplementary Information:

Supplementary Fig. 6. Description of battery connection. a, Photographs showing the magnetic LiPo battery connection interface using neodymium magnets. **b,** Battery life test for 40-mAh capacity.

Comment #2: “Fig. 4 shows different finger gestures used to control a robotic hand, but how was it possible to isolate the ring finger from the little finger in the final photo?”

Our response:

We appreciate this valuable comment. The identification of specific finger movement is determined by the unique combination of RMS intensities from all three p-NHE devices. As shown in Fig. 4e, these RMS data, generated from each finger movement, can be represented as a single data point in the xyz coordinates.

Further evident from the figure is that the data points tend to form a cluster for each finger movement, over repeated trials. Based on this finding, we defined three-dimensional classification boundaries for each finger movement (areas inside dotted ellipses) that would automatically determine where the next RMS data should fall within. While the type of muscle activations between the ring and the little fingers may appear similar, their RMS clusters occupy distinctively different spaces in the three-dimensional domain. The ability to distinguish the movements between these two fingers with a high accuracy is another example that demonstrates the versatility of p-NHE.

A new table, shown below, shows the average RMS values for each finger movement, which clearly shows the difference between each motion.

	Channel 1 (x10 ⁻⁴ V)	Channel 2 (x10 ⁻⁴ V)	Channel 3 (x10 ⁻⁴ V)
Open	0.48	0.31	0.29
Close	4.00	1.42	2.72
Thumb	0.44	0.45	2.19
Index	1.65	0.48	0.70
Middle	1.22	0.46	0.42
Ring	1.92	0.35	3.24
Little	1.26	0.42	0.71

Our modification to the manuscript:

For additional clarification, we add the following paragraph in Page 5:

“A 3D, three-channel RMS plot from three devices (Fig. 4e) shows seven distinctive clusters, generated by motions of individual fingers and hand gestures over repeated trials. A set of three p-NHEs, distributed over three different muscle groups, clearly captures different RMS values per each motion, as summarized in Supplementary Table 10. The resultant confusion matrix in Fig. 4f, generated by the developed classification algorithm, shows the overall accuracy of 98.6% for seven classes over ten different trials from multiple subjects.”

In addition, we add a new table (Supplementary Table 10) that summarizes the average RMS values from three devices:

Supplementary Table 10. Comparison of average RMS values from synchronized multi-device EMG recording. The table lists the RMS values calculated from Supplementary Fig. 23.

	Channel 1 (x10 ⁻⁴ V)	Channel 2 (x10 ⁻⁴ V)	Channel 3 (x10 ⁻⁴ V)
Open	0.48	0.31	0.29
Close	4.00	1.42	2.72

Thumb	0.44	0.45	2.19
Index	1.65	0.48	0.70
Middle	1.22	0.46	0.42
Ring	1.92	0.35	3.24
Little	1.26	0.42	0.71

Comment #3: “Finally, there are various minor grammatical errors or typos in the manuscript.”

Our response:

We thank the reviewer for pointing this out. We had thorough reviews with three different native speakers.

Our modification to the manuscript:

Three external people who are native speakers reviewed the manuscript, while fixing any grammatical errors and/or typos in the manuscript.

Reviewer #3 Remark:

This paper presents an additive manufacturing process to create multi-layer circuit boards for wearable devices. The overall approach is interesting and reasonably well executed.

Comment #1: “The authors do not contrast their approach to that from J. Rogers (Northwestern) or S. Xu (UCSD), whom have made multi-layer flexible circuit technology in the recent past, which includes the ability to mount commercial chips onto the developed substrates. The authors are strongly encouraged to explicitly compare their approach to these others, stating both the advantages and disadvantages.”

Our response:

We thank the reviewer for this valuable comment. In the modified Introduction, we now explicitly acknowledge the works by the Rogers and Xu groups as well as others that relate the fabrication of multi-layered flexible circuits. In the modified sentences, we reinforce that the multi-layered flexible electronics demonstrated by these groups have been successful at demonstrating functional skin-wearable applications. At the same time, we clearly define how our technology differs from those works based on conventional microfabrication methods.

We would like to emphasize that this work is the first demonstration of a fully printed, multilayer electronics including not only the circuits but also the electrodes and the stretchable connectors. We also modified the section to clearly emphasize our approach is advantageous to traditional CMOS processes because the entire device can be manufactured using a single printer, unlike the microfabrication where a cleanroom facility with multiple equipment are needed.

To help the reader better grasp the landscape in the manufacturing of flexible wearable electronics and how this work stands apart from other demonstration, we created below table to follow the Introduction.

Reference	Fabrication method	Sensor material	Circuit material	Sensor type	Acquisition mode
This work	All-printed	Graphene/PI on the elastomer	PI/Graphene/Ag/PI/Ag/PI on the elastomer	EMG	Synchronized, multi-device
[1] Rogers group	Microfabrication (Sensor)	Au/Cr on the adhesive silicone/elastomer	None	EMG	Single device
[2] Xu group	Casting (Sensor) Laser ablation (Circuit)	CNT and silbione	Four layers with a Cu/PI on the elastomer	EMG	Single device
[3] Yu group	Microfabrication (Sensor)	PI/IZO/Au/Cr/PI on the elastomer	None	Strain	Single device
[4] Yeo group	Printed (Sensor) Microfabrication (Circuit)	Ag/PI on the elastomer	PI/Cu/PI/Cu/PI on the elastomer	EOG	Single device
[5] Mercier group	Printed (Sensor)	Ag/AgCl on the polyester	Rigid circuit board	ECG and lactate	Single device
[6] Rogers group	Microfabrication (Sensor and circuit)	PI/Au/Cr/PI on the elastomer	Cu/PI/Cu on the adhesive	Temperature	Single device
[7] Xu group	Laser ablation (Sensor)	PI/Cu/transducer/Cu/PI on the elastomer	None	Blood pressure	Single device
[8] Xu group	Laser ablation (Sensor)	PI/Cu/transducer/Cu/PI on the elastomer	None	Ultrasonic probe	Single device
[9] Rogers group	Microfabrication (Sensor and circuit)	PI/Cu/PI/Cu/PI on the elastomer	PI/Cu/PI/Cu	Blood flow	Single device
[10] Rogers group	Microfabrication (Sensor)	PI/Cu/PI on the elastomer	None	Thermal	Single device

[1] Large-area MRI-compatible epidermal electronic interfaces for prosthetic control and cognitive monitoring, Nature Biomedical Engineering, 3, 194-205(2019)

[2] Three-dimensional integrated stretchable electronics, Nature Electronics, 1, 473–480(2018)

[3] Metal oxide semiconductor nanomembrane-based soft unnoticeable multifunctional electronics for wearable human-machine interfaces, Science Advances, 5, eaav9653 (2019)

- [4] Soft, wireless periocular wearable electronics for real-time detection of eye vergence in a virtual reality toward mobile eye therapies, *Science Advances*, 6, eaay1729 (2020)
- [5] A wearable chemical–electrophysiological hybrid biosensing system for real-time health and fitness monitoring, *Nature Communications*, 7, 11650 (2016)
- [6] Wireless, Battery-Free Epidermal Electronics for Continuous, Quantitative, Multimodal Thermal Characterization of Skin, *Small*, 14, 1803192 (2018)
- [7] Monitoring of the central blood pressure waveform via a conformal ultrasonic device, *Nature Biomedical Engineering*, 2, 687-695 (2018)
- [8] Stretchable ultrasonic transducer arrays for three-dimensional imaging on complex surfaces, *Science Advances*, 4, eaar3979 (2018)
- [9] Epidermal electronics for noninvasive, wireless, quantitative assessment of ventricular shunt function in patients with hydrocephalus, *Science Translational Medicine*, 10, eaat8437 (2018)
- [10] Flexible and Stretchable 3 ω Sensors for Thermal Characterization of Human Skin, *Advanced Functional Materials*, 27, 1701282 (2017)

Our modification to the manuscript:

Following sentences are added in the Introduction:

“Notable examples are the recent breakthroughs in the fabrication of thin, multi-layered flexible electronics with abilities to mechanically conform to human physiology as well as incorporate commercial electronic components for functional bioelectronic applications.³⁻⁵ While these platforms successfully demonstrated the utility of the well-defined, traditional CMOS processes toward unique wearable applications, the fabrication processes require the access to cleanroom facility, high-vacuum equipment, and dedicated personnel for maintenance. From this point of view, the ability to manufacture stretchable hybrid electronics entirely based on additive manufacturing methods is particularly attractive due to decreased material consumption, fast turnaround, scalable fabrication based on parallel printing, and, most importantly, the fact that only a single equipment is needed⁶. Table 1 summarizes the comparison of existing studies on multi-layered flexible electronics with this work using the printing method^{3-5, 7-13}.”

Table 1. Comparison of multi-layered sensor and circuit systems.

Reference	Fabrication method	Sensor materials	Circuit materials	Sensor type	Acquisition mode
This work	All-printed	Graphene/PI on the elastomer	PI/Graphene/Ag/PI/Ag/PI on the elastomer	EMG	Synchronized, multi-device
3	Microfabrication (Sensor)	Au/Cr on the adhesive silicone	None	EMG	Single device
4	Casting (Sensor) Laser ablation (Circuit)	CNT and Silbione	Four layers with a Cu/PI on the elastomer	EMG	Single device
5	Microfabrication (Sensor)	PI/IZO/Au/Cr/PI on the elastomer	None	Strain	Single device
7	Printed (Sensor) Microfabrication (Circuit)	Ag/PI on the elastomer	PI/Cu/PI/Cu/PI on the elastomer	EOG	Single device
8	Printed (Sensor)	Ag/AgCl on the polyester	Rigid circuit board	ECG and lactate	Single device
9	Microfabrication (Sensor and circuit)	PI/Au/Cr/PI on the elastomer	Cu/PI/Cu on the adhesive	Temperature	Single device
10	Laser ablation (Sensor)	PI/Cu/transducer/ Cu/PI on the elastomer	None	Blood pressure	Single device
11	Laser ablation (Sensor)	PI/Cu/transducer/ Cu/PI on the elastomer	None	Ultrasonic probe	Single device

12	Microfabrication (Sensor and circuit)	PI/Cu/PI/Cu/PI on the elastomer	PI/Cu/PI/Cu	Blood flow	Single device
13	Microfabrication (Sensor)	PI/Cu/PI on the elastomer	None	Thermal	Single device

Comment #2: “One of the biggest challenges facing flexible electronics in general - even those developed with commercially-available techniques, is reliability. The authors touch on this on slightly in the manuscript by demonstrating functional operation during bending. However, the results are not convincing. The paper would be strengthened considerably if results for yield (i.e. how many of the fabricated devices had full or partial functionality), and more repeated stress tests were reported.”

Our response:

We appreciate the reviewer for requesting additional information on device reliability and manufacturing yield. In our response, we additionally measured the functional operation for the integrated system (Supplementary Fig. 16). The EMG data and SNR before and after 100 bending cycles were summarized. The negligible changes in the quality of EMG are convincing evidence that the printed system will remain functional even after being folded 100 times in 180°, which is already an extreme case.

Supplementary Fig. 16. Device functionality for the integrated EMG system. Forearm EMG characteristics of the integrated system before (left) and after (right) 100 bending.

This new result, along with the previous reliability data presented in Fig. 3g, Supplementary Fig. 10, 11, and 15 have been tabulated as quantitative values to help readers have the better overall understanding on the mechanical reliability of p-NHE as an aggregate system. This information has been summarized as Supplementary Table 4.

Supplementary Table. 4a

	Electrodes		Circuit
	Bending	Stretching	Bending
R/R ₀ (%)	0.1	0.2	0.1

Supplementary Table. 4b

b	Integrated system
	Bending
Acceleration (%)	x: 0.3, y: 1.2, z: 1.1
SNR (%)	2.0

As for the yield of devices, we have reviewed the device yields at each process, including the printing, soldering, and integration, which is summarized as a table:

	Printing		Soldering	Integration
	Sensor	Circuit		
Yield (%) (Operations/Trials)	93 (9.3/10)	90 (9/10)	100 (9/9)	67 (6/9)

The optimized printing and soldering processes provide the high yield of above 90% where most failure modes are due to human error during the sample preparation or machine operation. However, since the integration of the sensor and the circuit is conducted manually, the yield is further reduced to 67%. We anticipate that this yield will also be improved if an automated process is implemented.

Our modification to the manuscript:

For additional clarification, we add the following paragraph in Page 5 of the manuscript:

“For the reliability study of the integrated p-NHE, we conducted EMG recording and SNR comparison between before and after 100 bending cycles (Supplementary Fig. 16 and Supplementary Table 4b), which shows a negligible change of the EMG quality. In addition, the fabrication yield was calculated based on each process, including the printing, soldering, and integration (Supplementary Table 7). The optimized printing and soldering processes provide the high yield of above 90%, while the final integration of sensor and circuit reduces the yield. Once an automated device assembly is implemented, then the overall yield will be significantly improved.”

In addition, we add the following figure and tables to the Supplementary Information.

Supplementary Fig. 16. Device functionality for the integrated EMG system. Forearm EMG characteristics of the integrated system before (left) and after (right) 100 bending.

Supplementary Table 4. Summary for the reliability of p-NHE. **a**, Structural reliability of the electrode and the circuit calculated from Fig. 3g and Supplementary Figs 10, 11. **b**, Functional reliability of the integrated system calculated from Supplementary Figs 15, 16.

a

	Electrodes		Circuit
	Bending (% change after 100 cycles)	Stretching (% change after 100 cycles)	Bending (% change after 100 cycles)
R/R ₀	0.1	0.2	0.1

b

	Integrated system
	Bending (% change after 100 cycles)
Acceleration	x: 0.3, y: 1.2, z: 1.1
EMG SNR	2.0

Supplementary Table 7. The fabrication yield at each process. The table indicates the yields for printing, soldering, and integration process with the optimized conditions.

	Printing		Soldering	Integration
	Sensor	Circuit		
Yield (%) (Operations/Trials)	93 (9.3/10)	90 (9/10)	100 (9/9)	67 (6/9)

Comment #3: “Lastly, the authors do not properly describe how this system is powered. More details in this regard are needed.”

Our response:

We thank the reviewer for pointing this out. In fact, the electronic system is powered by a lithium-ion polymer battery (40 mAh; DTP301120, Shenzhen Data Power Technology) as shown in the figure below. The miniaturized neodymium magnets facilitate an easy and secure connection of the battery, which are placed on both the circuit and the battery. When the battery is attached, it is indicated by the green LED, while the blue LED shows the connection of Bluetooth between

the device and external tablet. In this figure, we added an experimental result that shows the battery (40 mAh) operation time. Depending on the device applications, data sampling rate and battery capacity can be changed to provide sufficient power.

Our modification to the manuscript:

The following description about the battery is added in Page 3 of the Manuscript:

“The integrated p-NHE is powered by a miniaturized lithium-ion polymer battery (40 mAh capacity; DTP301120, Shenzhen Data Power Technology). The battery’s two terminals and the circuit’s power pads are soldered with small neodymium magnets for a guided battery connection. The details for battery connection and power efficiency are described in Supplementary Fig. 6.”

In addition, the following figure is added in the Supplementary Information:

Supplementary Fig. 6. Description of battery connection. a, Photographs showing the magnetic LiPo battery connection interface using neodymium magnets. **b,** Battery life test for 40-mAh capacity.

Reviewers' Comments:

Reviewer #2:

None

Reviewer #3:

Remarks to the Author:

It appears that the main contribution of this work is moving from a lithography- or laser-defined subtractive manufacturing process to a printed additive-manufacturing process. Not being an expert in this specific area, I'm not sure this is a sufficient enough advance to warrant publication in a Nature sub journal - I leave that to the editor and the other reviewer(s). Beyond this point, the overall engineering work here is well done and my initial questions have been addressed.